# Developing a core outcome set for interventions to improve discharge from mental health inpatient services: a survey, Delphi and consensus meeting with key stakeholder groups

Natasha Tyler  ,[1] Nicola Wright,[2] Andrew Grundy,[2] Justin Waring[1,3]

¹NIHR Greater Manchester Patient Safety Translational Research Centre, The University of Manchester, Manchester, UK
²Health Sciences, University of Nottingham, Nottingham, UK
³Health Services Management Centre, University of Birmingham, Birmingham, UK

**Correspondence to**
Dr Natasha Tyler;
natasha.tyler@manchester.ac.uk

## ABSTRACT

**Objective** To develop a core set of outcomes to be used in all future studies into discharge from acute mental health services to increase homogeneity of outcome reporting.

**Design** We used a cross-sectional online survey with qualitative responses to derive a comprehensive list of outcomes, followed by two online Delphi rounds and a face-to-face consensus meeting.

**Setting** The setting the core outcome set applies to is acute adult mental health.

**Participants** Participants were recruited from five stakeholder groups: service users, families and carers, researchers, healthcare professionals and policymakers.

**Interventions** The core outcome set is intended for all interventions that aim to improve discharge from acute mental health services to the community.

**Results** Ninety-three participants in total completed the questionnaire, 69 in Delphi round 1 and 68 in round 2, with relatively even representation of groups. Eleven participants attended the consensus meeting. Service users, healthcare professionals, researchers, carers/families and end-users of research agreed on a four-item core outcome set: readmission, suicide completed, service user-reported psychological distress and quality of life.

**Conclusion** Implementation of the core outcome set in future trials research will provide a framework to achieve standardisation, facilitate selection of outcome measures, allow between-study comparisons and ultimately enhance the relevance of trial or research findings to healthcare professionals, researchers, policymakers and service users.

## BACKGROUND

Care transitions (when patient care is transferred from one team, department or organisation to another) are widely recognised as a vulnerable and high-risk stage in the care pathway.[1–3] Safety issues may be intensified in acute mental health services, where care transitions are described as chaotic.[3] For example, suicide risk increases postdischarge from acute mental health services.[4 5] A growing body of research describes these

## Strengths and limitations of this study

► This is the first initiative to reduce heterogeneity in outcome reporting for interventions that improve discharge from acute mental health services.
► A high level of consensus among 69 service users, families/carers, healthcare professionals, researchers and policymakers was achieved.
► Core Outcome Set-Standards for Reporting guidelines were followed.
► Although the stakeholder group included international researchers, service users and healthcare professionals were recruited only from the UK.
► Not all of the participants who contributed online attended the face-to-face meeting, whereby the core outcome set was reduced considerably.

risks either directly in terms of identified 'safety' events or indirectly in terms of broader 'problems', including, for example, treatment non-adherence, inappropriate readmissions, increased risk of self-injury or suicide attempts.[3 6–8]

Internationally, researchers have attempted to find solutions to the problems or threats to safety associated with discharge from acute mental health services by developing interventions that aim to improve different aspects of discharge planning, transitions, continuity of care and follow-up care.[9] Some interventions aim to improve discharge by introducing new roles, for example, a discharge coordinator.[10] Others focus on increasing contact between clinical staff and service users, for example, using letters or telephone follow-up.[7 11 12] Many 'successful' interventions in reducing readmission, bridged the boundaries between ward and community by providing types of ward-based care in the community[13 14] or where community teams lead discharge planning on the wards.[15]

There has been little attempt to compare these diverse interventions. Existing reviews have included either a narrow range of studies addressing a single outcome or focus on a specific time frame in an attempt to synthesise results.[8 16] Comparison and meta-synthesis of effectiveness of interventions have reported limited success. Across the papers included in our systematic review and those by other researchers,[1 16] variation in the outcomes reported is substantial. This limits between study comparability and delays advancement in evidence collection. Furthermore, outcomes in these trials were not necessarily representative of the measures that service users would consider important at discharge. Both matters can potentially be addressed with the development of a 'core outcome set', defined as 'an agreed, standardised collection of outcomes which should be measured and reported, as a minimum, in all trials for a specific clinical area'.[17]

The development and use of 'core outcome sets' have been endorsed as a means to reduce outcome heterogeneity in research and to increase the relevance of research through the involvement of key stakeholders in its development.[18] There is an emerging body of the literature highlighting the difficulties of defining and assessing outcomes in a mental health population.[19] There is also evidence of a lack of agreement among key groups about what should be measured and in what capacity and an evident tension between the population health perspective and provision of individualised care.[16 19] One aforementioned previous review identified the need for consensus on outcome definitions in discharge planning interventions.[16] Similarly, a recent Kings Fund report suggested broader consensus on the outcomes that matter is imperative for advancement.[19] Therefore, generating agreement among healthcare professionals, service users, policymakers and researchers is a difficult but imperative task, to enable the useful direction of healthcare services.[19] The difficulties are further exemplified when applied to care transitions, a multiagency, multistage, complex period of the care pathway.[3 20] This paper outlines the development of a core outcome set for research of interventions to improve discharge from acute mental health wards to the community.

The objective of this study was to obtain international consensus on a set of core outcome measures to be reported in all interventions intended to improve discharge from mental health inpatient services.

## METHODS
### Study overview
The scope of the core outcome set was defined according to the criteria recommended by Core Outcome Measures in Effectiveness Trials (COMET).[21] The study was prospectively registered with the COMET initiative (1276). The health condition was functional mental health coinnditions (conditions other than dementia and includes severe mental illness such as schizophrenia). The population was adults aged 18–65, the intervention was any

interventions that aimed to improve discharge from an acute mental health setting to the community. The core outcome set was developed using four stages, including service users and healthcare professionals at each stage: (1) a long list of outcomes was generated through a systematic review[1] and qualitative survey; (2) the resulting long outcome list was used to populate an online Delphi process (two rounds) and (3) the results of the Delphi survey were appraised at a consensus meeting and the final core outcome set was established. The process included a series of core research team meetings at every stage, the team comprised of a researcher and core outcome set developer, an associate professor in mental health and mental health nurse, a researcher and expert by lived experience of acute services and an expert in patient safety. Participants did not fit into distinct homogeneous groups, for example, mental health professionals were sometimes also past service users or family members of service users. Similarly, researchers had personal experience of inpatient mental health services. Therefore, wherever possible we considered the group as whole and tried not to compare categories.

### Participants
Participants were recruited in a number of ways from December 2018 to January 2019.

Academic researchers were recruited if their research had been included in our systematic review or if they were known researchers in the field identified by the team. End-users of research (policymakers, Nongovernmental organisations, National Health Service (NHS) management, commissioners, advocates, and so on) were recruited via searching for publicly available contact details or using our team's professional networks or social media. Service users and healthcare professionals were recruited through social media. Twitter was nominated as the primary platform for recruitment due to its ability to reach into the specific communities of interest we required: mental health professionals, service users and families/carers. Using social media has been reported as a cost-effective and efficient way to recruit those from potentially stigmatised groups.[22] Further, the peer network structures of social media platforms enable users to recruit other users through sharing links within their networks.

The same participant group was used throughout the iterative research process, therefore, in order to reduce attrition, those who dropped out in early rounds were invited to rejoin the panel in subsequent rounds. Participants were recruited for the consensus meeting during the Delphi, UK participants were asked to indicate whether they would be interested in a face-to-face meeting. We invited a random sample of interested participants to attend, which ensured the representative of the stakeholder groups. If a participant declined the invite, a similarly matched participant was invited from the Delphi panel principally or the teams wider network.

## Stage 1: gathering information

In addition to the outcomes extracted from the systematic review,[1] outcomes of importance to each stakeholder group were identified through qualitative surveys. For the main body of the questionnaire, we used open questions that were developed to elicit potential additional outcomes. The questions were loosely modelled on questions developed for a large-scale outcome generation study for a depression core outcome set that were developed with service users and healthcare professionals.[23] The question format was mirrored but adapted for a mental health discharge theme. The views of a patient and public involvement (PPI) group sought to confirm the appropriateness of questions and instructions (n=5).

After reading a participant information sheet and giving informed consent (by ticking a box), participants selected their stakeholder group(s) and watched a video that describes core outcome sets to non-experts. All participants were then presented with four open-ended questions relating to safe and effective discharge (see online supplementary file 1). Participants were later presented with three to five questions specifically developed for their stakeholder group, online supplementary file 1 outlines all of the questions. If a participant was a member of more than one group, they answered questions relevant to multiple groups. Participants also answered a number of demographic questions: years of experience, country of residence, area of UK (if applicable), gender, age and email address for follow-up. The round was open for 6 weeks beginning 1o December 2018.

Qualitative data were coded to identify outcomes and thematically synthesised.[24] This involved line-by-line coding of text and development of descriptive themes, the final stage involved generating analytical themes, which were converted into potential outcomes where applicable. Outcomes were identified both indirectly, by extrapolating from service users' experiences (eg, What would make discharge from an acute mental health ward safe in your opinion?), and directly, by asking specifically about outcomes (eg, Can you think of any important outcomes to measure in research assessing discharge interventions?).

Outcomes from the systematic review[1] and qualitative surveys were combined to generate a long list of outcomes. This list, along with relevant quotes from the qualitative data, was discussed by the core research team in a structured meeting. Each outcome was considered in turn and each member had the opportunity to present arguments for or against inclusion. For each outcome, the group decided whether it should be a stand-alone outcome, combined with other codes of a similar thematic nature or removed from the process due to being of limited importance for a core outcome set. For example, we agreed to merge closely related items (eg, *family relations* and *quality of interpersonal relationships*) and to exclude outcomes considered to be of limited importance (eg, specific to a specialised area of care: *autistic life*; or intervention *antipsychotic politherapy*). Unless there was a unanimous decision to merge or remove an outcome, it remained as a stand-alone outcome. The group decisions about each outcome are documented in online supplementary file 1.

## Stage 2: Delphi survey

The Delphi technique is a research method aimed at generating consensus. It solicits opinions from stakeholder groups in an iterative process of answering questions. After each round, the responses are summarised and redistributed for discussion in the next round. We chose to have two rounds of Delphi in this study. The final outcome list that was decided on after the group discussion in stage 1 was used to develop the first Delphi questionnaire. Any outcomes without consensus after the first round were represented in round 2. The outcome list and instructions for the questionnaires were reviewed for face validity, understanding and acceptability by a PPI group (n=5) and modified according to the feedback.

A link to the survey was sent via email. Each round remained open for 14 days and participants received two follow-up reminder emails. Round 1 was open from late February 2019 to early March 2019, round 2 was from late March 2019 to early April 2019. We ran the Delphi survey manually using Qualtrics: a secure online hosting platform.[25] In each round, participants were asked whether the items should become part of a core outcome set. A 7-point Likert scale was used, described as: strongly agree (7), agree (6), slightly agree (5), neither agree nor disagree (4), slightly disagree (3), disagree (2) and strongly disagree (1). There is no definitive research indicating the optimal number of points to have on a Likert scale but scales between 5 and 9 points have been suggested as having the best reliability, so we chose a 7-point scale.[26] There was a free-text comment box and participants were encouraged to provide comments that would be fed back anonymously to the group. Participants could suggest additional outcomes at the end of round 1, which were reviewed by the core research team. Any outcome not already represented was added to round 2.

In round 2, median group scores for each outcome and anonymous comments for and against from the previous round were presented and participants were asked to reflect on the information presented and score each outcome again. The percentage of participant agreement with each outcome on a scale of 1–7 was calculated from the scores obtained during round 1 and again in round 2.

Literature suggests that consensus levels should be set a priori at a minimum of 70 per cent.[21 27] We unanimously chose a 75% consensus level, slightly higher than the minimum to increase sensitivity, but to still allow for a varied pool of applicable outcomes given the tension in the literature around disagreement between service user, health professional and policy-maker opinions of mental health outcomes.[19] Consensus criteria were defined a priori: outcomes scored as agree or strongly agree (6 or 7) by 75% or more of the group reached consensus for inclusion and were included in the provisional core

outcome set. Outcomes scored as disagree or strongly disagree (1 or 2) by 75% or more were defined as having reached consensus for exclusion and were excluded. Outcomes not fulfilling criteria for consensus inclusion or exclusion were defined as not having reached consensus and were represented in round 2.

As no outcomes met the original criteria for having reached consensus for exclusion after round 1, it was agreed by the research team to redefine the criteria for having reached consensus for exclusion if 50% or less of participants scored the item as strongly agree or agree (6 or 7). Reducing exclusion criteria after round 1 has been used effectively in past core outcome set research.[28]

### Stage 3: consensus meeting

The results of the Delphi survey were presented at a consensus meeting. The main goal of the consensus meeting was to decide which items will be included in the final core outcome set. This was chaired by an independent researcher with expertise in consensus methodology, and who was not a member of the core research team. Participants were sampled to achieve a balanced representation of service users, healthcare professionals, researchers and end-users of the research. We aimed to have a small representative group between 9 and 12 to enable meaningful small group discussions, similar to consensus meetings chaired by the facilitator in other fields.[28 29] International participation was restricted because of budgetary constraints.

The format of the consensus meeting comprised of (1) a short overview of the study and (2) a summary of the Delphi results sorted by stakeholder group, beginning with the outcomes that met consensus.[30] Outcomes identified in rounds 1 and 2 of the Delphi as having reached consensus for inclusion were presented first. Participants were asked if there were any fundamental reasons why these should not be included in the core outcome set. Divergent views were actively sought and the chair ensured everyone had opportunity to participate in discussions before voting commenced. Outcomes from the preliminary core outcome set were discussed in terms of feasibility and voted on. Voting was conducted anonymously using cards in an envelope with bivariate response options (include/exclude). Voting and consensus criteria followed the same format as in the Delphi (75% for inclusion). Results were presented after the voting of all outcomes had finished. Outcomes deemed to be having reached consensus for exclusion or with no consensus in the Delphi were reviewed and participants were asked if there were any fundamental reasons why these should be included in the core outcome set. Individual outcomes were discussed only if proposed as being important by a meeting participant. Outcomes meeting criteria for consensus were included in the core outcome set; all other items excluded. The meeting finished with the presentation and a final review and discussion of the core outcome set.

### Patient and public involvement

Five patient representatives worked with researchers to develop the online questionnaires. Patients were represented alongside professionals and researchers in the consensus panel. One member of the research team (and coauthor) is an expert by lived experience and was involved in all design and analysis decisions.

### Registration

Our findings are reported in line with the Core Outcome Set-Standards for Reporting (COS-STAR) guidance.[30] The study was prospectively registered with the COMET initiative (1276).[30]

## RESULTS
### Stage 1: information gathering

Our systematic review has been described in detail elsewhere.[1] In summary, 69 outcome categories were identified from 45 studies. Ninety-three participants in total, from 12 countries, completed the information gathering questionnaire. However, as aforementioned, many identified with more than one stakeholder group, therefore, we do not have absolute homogenous stakeholder group numbers, 27 identified as service users, 17 family/carers, 39 healthcare professionals, 15 end-user of research and 37 researchers. Online supplementary file 1 presents participants demographics. Qualitative questionnaires revealed an additional 45 outcomes that were not identified in the literature (eg, outcomes concerning involvement in discharge planning, see online supplementary file 1). After discussion within the research team, 82 standardised outcome terms were taken forward into the Delphi process; 19 outcomes were combined/collapsed and 13 were removed, see online supplementary file 1.

### Stage 2: Delphi process

Sixty-nine participants completed round 1 of the Delphi (22 service users, families and carers, 26 researchers and 21 healthcare professionals and decision-makers) and 68 participants completed round 2 (30 researchers, 18 service users and families and 20 healthcare professionals and decision-makers). While five participants dropped out after round 1, four participants joined the panel in round 2 (these individuals participated in the qualitative questionnaire but not round 1). There was 1.4% attrition between rounds 1 and 2 of the Delphi. Seven additional outcomes were proposed by participants during round 1, of which two were added into round 2 after a core team discussion. The full list of Delphi items is available in online supplementary file 1.

After round 1, 14 outcomes met the criteria for consensus inclusion (75% or more agreed/strongly agree with that outcome, see table 1). Twenty outcomes met the revised criteria for having reached consensus for exclusion (50% or less of participants agreed/strongly agreed with that outcome). Forty-eight outcomes did not meet consensus criteria for inclusion or exclusion and were

**Table 1** The preliminary core outcome set at the end of the online Delphi

| | Percentage agreement | Percentage disagreement | Median | Researchers (%) | Service users and families (%) | HCPs and DMs (%) |
|---|---|---|---|---|---|---|
| Service user involvement in discharge planning (including feeling listened to) | 87 | 4 | 7 | 65 | 100 | 95 |
| Functioning (health, social, etc) | 83 | 3 | 6 | 69 | 100 | 81 |
| Mental health and illness (symptom/psychological distress) | 83 | 3 | 6 | 73 | 91 | 86 |
| Personal recovery | 82 | 1 | 6 | 75 | 86 | 86 |
| Service user understanding of discharge plan | 81 | 3 | 6 | 65 | 91 | 86 |
| Quality of life | 81 | 1 | 6.5 | 65 | 90 | 86 |
| Suicide completed | 80 | 4 | 7 | 80 | 90 | 68 |
| Readmission | 80 | 6 | 6 | 77 | 77 | 86 |
| Service user involvement in decision-making (shared decision-making) | 77 | 4 | 7 | 50 | 95 | 86 |
| Service user satisfaction with information provision at discharge (eg, regarding medication, risk, crisis planning) | 77 | 6 | 6 | 65 | 86 | 81 |
| Service user knowledge of how to access community support (ie, in an emergency) | 77 | 3 | 6 | 58 | 91 | 86 |
| Recurrence (ie, relapse) | 75 | 1 | 6 | 58 | 91 | 76 |
| Suicide attempted | 75 | 4 | 6 | 62 | 86 | 81 |
| Discharge to appropriate accommodation | 75 | 3 | 6 | 69 | 91 | 67 |
| Meaningful activity (included in round 2) | | | | 73 | 80 | 79 |

HCPs (Healthcare Professionals), DMs (Decision Makers/End users of research)

represented to the group in round 2. Therefore, 50 outcomes were presented in round 2, only one outcome met the criteria for consensus after this round: meaningful activity. No outcomes met criteria for exclusion and 49 did not meet consensus. Online supplementary file 1 shows the consensus levels for each outcome in each round.

### Stage 3: consensus meeting
Eleven participants attended the consensus meeting, as in previous rounds these categories were not exclusive, six participants were researchers, three identified as service users, three as healthcare professionals and three end-users of research, see table 2. Table 3 shows the quantitative results of the meeting.

The preliminary 15-item core outcome set was considered individually and discussions indicated that many of the outcomes were elements of an ideal discharge, and process outcomes/variables, but probably not measurable outcomes that should be included in a core outcome set. After these discussions and independent and anonymous voting, five items no longer met consensus criteria for

**Table 2** Participants who attended consensus meeting

| Participant number | Researcher | Service user | Healthcare professional | End-user research |
|---|---|---|---|---|
| 1 | X | | | |
| 2 | X | | | |
| 3 | X | | | |
| 4 | X | | X | |
| 5 | X | X | | |
| 6 | X | | | X |
| 7 | | | X | |
| 8 | | | X | X |
| 9 | | X | | |
| 10 | | X | | |
| 11 | | | | X |
| Total | 6 | 3 | 3 | 3 |

**Table 3** Outcomes of consensus meeting, levels of consensus in anonymous voting

| | Include | Exclude | Percentage |
|---|---|---|---|
| Readmission | 10 | 1 | 91 |
| Service user reported psychological distress | 11 | 0 | 100 |
| Suicide completed | 9 | 2 | 82 |
| QoL | 9 | 2 | 82 |
| Reoccurrence | 4 | 7 | 36 |
| Mental health and illness | 8 | 3 | 73 |
| Service user involvement in decision-making | 7 | 4 | 64 |
| Personal recovery | 6 | 5 | 55 |
| Meaningful activity | 1 | 10 | 9 |
| Functioning | 1 | 10 | 9 |
| Clinician-reported mental health | 5 | 6 | 45 |
| Service user satisfaction with information provision at discharge | 3 | 8 | 27 |
| Service user understanding of the discharge plan | 3 | 8 | 27 |
| Suicide attempted | 3 | 8 | 27 |
| Service user involvement in discharge planning | 6 | 5 | 55 |
| Knowledge of how to access support in a crisis | 5 | 6 | 45 |
| Discharge to appropriate accommodation | 0 | 11 | 0 |

QoL, quality of life.

**Table 4** The final core outcome set

| | Final core outcome set |
|---|---|
| 1 | Readmission |
| 2 | Quality of life |
| 3 | Suicide completed |
| 4 | Service user-reported psychological distress |

inclusion. First, 'service user involvement' in discharge planning and the associated items 'service user understanding of discharge plan', 'service user involvement in decision-making', 'service user satisfaction with information provision at discharge' and 'service user knowledge of how to access community support'. There was a discussion that these are very important elements of a successful discharge, but not core outcomes due to issues surrounding validity and meaning.

'Mental health and illness' was initially close to consensus with 73% consensus to include, however those that chose to exclude found it to be too vague, and articulated that they were most interested in measuring acute psychological distress, rather than mental health and illness. The service user representatives in the group interpreted 'recovery' to mean a complete amelioration of symptoms and even when in 'recovery' individuals described continuing to experience distress and difficulties with their mental health. We chose to therefore separate the broader mental health and illness outcome into self-reported psychological distress and clinician-reported mental health. The granular outcome of self-reported

psychological distress resulted in 100% consensus to include. On the contrary, clinician-reported mental health did not meet consensus criteria (45%). Similar discussions happened around the recurrence (relapse) outcome, whereby its inclusion in a core outcome set, would ultimately necessitate buy-in to criteria model, which suggested that mental health problems could and should be completely resolved.

Discussions around the 'suicide attempted' outcome indicated that participants felt that suicide attempts or self-harm had diverse motivations and definitions and they discussed the issues of delineating the boundaries of self-harm and suicide attempts and how this is documented. After the consensus meeting, this outcome no longer meets consensus criteria to include. Discussions surrounding personal recovery, functioning and meaningful activity indicated that participants considered these outcomes too vague and subjective to be a component of a core outcome set. There was consensus to exclude meaningful activity and recovery, and no consensus to include personal recovery. There was consensus to exclude discharge to appropriate accommodation, discussion indicated this was primarily because this spanned the health and social care boundaries and may not be applicable to every intervention.

On completion of the meeting, only four outcomes met consensus criteria for inclusion, see table 4. A core outcome set of four was agreed, participants agreed that the following should be included: readmission, quality of life, suicide completed and service user-reported psychological distress. Readmission was the most frequently used outcome in past research, and despite limitations, participants felt it was one of the only proxy measures of appropriate discharge. Quality of life and psychological distress were considered important ways of quantitatively assessing the psychosocial elements of discharge; which are of primary importance. Suicide completed was considered rare but imperative data to capture given the research highlighting the relationship between acute mental health discharge and suicide highlighted by a growing body of literature.[5 31] Figure 1 shows the process undertaken to reach the core outcome set.

## DISCUSSION

This study provides the first international consensus on outcomes for intervention studies concerning discharge from an acute adult mental health inpatient setting. We could not identify any other published core outcome

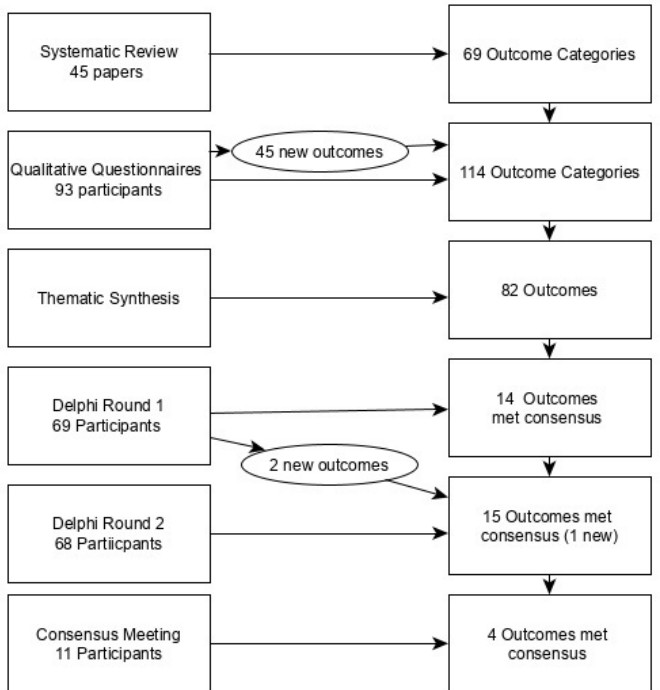

**Figure 1** Process of reducing the outcomes to a set of four core outcomes.

sets for interventions concerning discharge from acute mental health services. Moreover, there are very few core outcome sets for mental health, despite recommendations for consensus in the literature.[16 19] All the included outcomes were agreed by more than 75% of a group of relatively equally represented service users and family/carers, healthcare professionals, researchers and end-user of research using consensus methods. We recommend that all future research studies evaluating interventions for discharge from acute adult mental health settings use this core outcome set as a framework for outcome selection, to compliment, rather than replace any other outcomes that are relevant to their research question. As discharge from acute services is a particularly challenging period for those experiencing mental health problems,[3 31] it is important to understand what interventions work and more specifically which elements of an intervention improve which particular outcomes. This core outcome set provides a framework for between-study comparison, ultimately enabling researchers to articulate the theory of change that underpins interventions.

In our systematic review,[1] we identified 22 studies that reported readmission rates as an outcome, yet almost all of them captured this in different ways: some used self-report data, some clinical case notes or some retrospective administrative data, others used case manager's reports. In addition, the time markers were variable, some used country-specific time markers in line with policy such as 28 days in the UK,[32] while others chose a series of time markers such as within 1 month, 3 months and 6 months, but the time markers were rarely directly comparable. Similarly, six studies measured quality of life but only

two used the same measurement instrument (Lehman's quality of life).[15 33] In the current study, we have developed consensus that quality of life and readmission are important and feasible to measure, robust recommendations of how best to measure these are now needed.

There were some unexpected exclusions in the core outcome set, for example, mental health symptoms and treatment adherence were frequently used in past research,[1] but not included in the core outcome set. In the background of this paper, we described the recent Kings Fund report that suggested generating agreement among healthcare professionals, service users, policymakers and researchers is a difficult but imperative task.[19] Our work reiterates these findings, and the small four-item core outcome set represents the only outcomes that are unanimously agreed on, despite so many outcomes being of upmost importance to service users and families.

This research has further highlighted the importance of shared decision-making and service user and family involvement to all stakeholder groups.[34]

This study indicates an impending desire to assess service user satisfaction and involvement in the process. While such outcomes were excluded in later stages of this research, it does not reduce their prospective importance in discharge interventions or provision of care at discharge. The five most agreed on elements of service user involvement and satisfaction in discharge were: service user involvement in discharge planning; service user understanding of discharge plan; service user involvement in decision-making; service user satisfaction with information provision at discharge and service user knowledge of how to access community support. Policymakers and healthcare management might consider measuring these five things in local-level initiatives as overriding principles of care to ensure they are not missing from care provision.

Research highlighting the importance of involving service users in mental healthcare planning is emerging, along with measures of such activity. Therefore, we suggest that future research could include a service user-reported outcome measure of involvement alongside the four-item core outcome set and any other chosen measures. This could be measured in an existing instrument of service user involvement care planning in mental health, such as the EQUIP PROM (patient reported outcome measure).[34] The six outcomes described above can also be presented as self-reported Likert measure of service user involvement in discharge planning (see online supplementary file 1). These six items are developed from the synthesis of academic literature, qualitative questionnaires and met criteria for consensus among experts in round 2, so from a psychometric perspective would arguably meet initial face and content validity criteria.[35]

The difficulties of developing a mental health core outcome set were further epitomised when applied to care transitions: a service-level (rather than specific clinical population) multiagency, multistage, complex period of the care pathway.[3 20] Generating a set of meaningful

applicable outcomes that span primary and secondary care, across multiple physical locations, that are relevant for every service user was imperative. For example, a great deal of past literature focuses on housing interventions,[36–38] and while housing is a significant safety issue at discharge, it is not necessarily relevant to all service users. This multiagency and multimorbidity complexity was arguably one factor that resulted in the small set of generic outcomes that arguably differs from narrowly defined clinical core outcome set reported in the past literature of many more outcomes.[39 40]

This study had several strengths. Our method is based on recommendations from an international panel of experts.[21] Inclusion of service users and healthcare professionals at every stage ensured that outcomes in the final core set embody shared priorities. The comprehensive and laborious long-list process ensured that all potential outcomes were considered in the course of the consensus process. However, there were some limitations to our study. The research was only conducted in English, due to budgetary constraints, although our online rounds included participants from 12 countries. Furthermore, in many consensus meetings, additional outcomes are often added, the method infrequently serves as means of reducing the number of outcomes included in the preliminary core outcome set from the Delphi.[28] However, in our case, we found that the group did not agree with many of the outcomes and it was reduced to a very small COS (core outcome set) of four items. This is beneficial in some ways, as we hope it is easier for researchers to operationalise a four-item core outcome set.

The use of outcomes in mental health research and service is becoming more contested in terms of what is meaningful and effective, it could be argued that core outcome sets are less applicable to mental health populations than general health populations, given the complexity of mental health problems and the subjectivity of measuring it. However, as core outcome sets are relatively uncommon in mental health, we believe (similar to other clinical populations) a small, agreed, feasible set of core outcomes will facilitate between study comparability and advancement in evidence collection.[17 21]

### Future directions

Development of this core outcome set involved the participation of stakeholders from 12 different countries; (primarily researchers) however, we recommend that further work should be undertaken to validate this core outcome set more widely, particularly in non-English speaking populations. The two of the final four outcomes and many of the preliminary 15 outcomes to emerge from the Delphi are not necessarily specific to mental healthcare transitions. Some outcomes are comparable to a similar core outcome set for care transitions of adolescents and young adults with special healthcare needs.[41] Future research may consider a 'transitions of care' core outcome set, to reduce the number of similar core outcome sets.

Another key priority to make this core outcome set operationalised is to agree on measurement criteria using the COSMIN (COnsensus-based Standards for the selection of health status Measurement INstruments) guidelines.[42] We conducted some preliminary questionnaires with the Delphi panel to produce preliminary measurement recommendations, however there was very little agreement among panellists (see online supplementary file 1).[42] The recommended measures by the panel were Kessler Psychological Distress and Recovery Quality of Life within 1 month of discharge.[43 44] Readmission and suicide completed rates were recommended to be captured within 28 days of discharge using retrospective review of administrative data. However, these are only preliminary recommendations and we highly recommend a future study following COSMIN guidelines.

### CONCLUSION

The four outcomes included in our outcome set represent the consensus opinion of a group of service users, healthcare professionals and international researchers and address an unmet necessity: assisting researchers in the design, implementation and reporting of interventions that aim to improve discharge from acute mental health settings. Ultimately, application of this core outcome set will enhance the relevance of future interventions to healthcare professionals, the research community and service users. If used, the core outcome set could provide more evidenced-based interventions, underpinned by theory of change outlining the relationships between the component of the intervention and the outcome it should improve,[1 45] which should increase service user safety at this distressing time period.

**Acknowledgements** We would like to thank the patient and public involvement group for their help in designing the questionnaires. The following participants chose to be acknowledged for their participation and the remainder chose to remain anonymous:Amy Clowes, Andrew Gallivan, Anna Hegedus, Anna Willis, Bernd Puschner, Cara Sturgess, Cassandra Lovelock, Cheryl Forchuk RN PhD, Colette Ramsey, David Cochrane, David Smelson, David Smith, Deb Smith, Dr Leah Quinlivan, Dr Nicola Clibbens, Dr Sarah Markham, Dr.Pucciarelli Gianluca, PhD, E Thomas, Emma Ward, Hameed Khan, Hazel Nash, Helene Provencher, Jan Hutchinson, Jane McNeice, Jean Nicholls, Justin Scanlan, Kyri Gregoriou, Law YikWa, Liz Monaghan, Mark S. Smith, Mark Trewin, Matthew Mckenzie, Michael P. Hengartner, Mohammad Ghadirivasfi, Mrs Z Burns, Philippe Golay, Phyllis Solomon, Ph.D., Prof Roger Webb, Rebecca Musgrove, Rebecca Stack, Sarah, Sarah Gunn, Sarah Markham, Shaun Thomson, Umar Kankiya, Usha Sambamoorthi, Zoe Swithenbank.

**Contributors** NT conceived the design of the study; conducted the literature search, meta-synthesis and Delphi; analysed the data and drafted the majority of the manuscript. JW and NW contributed equally and provided oversight of the study design, analysed and synthesised the data and contributed significantly to the drafting of the manuscript. AG analysed and synthesised data, provided an expert by lived experience opinion on decisions made in regards to wording, study design, patient involvement and also contributed adaptations to the manuscript.

**Funding** This research is funded by the NIHR Greater Manchester Patient Safety Translational Research Centre, they had no role in the writing of manuscript or decision to submit.

**Competing interests** None declared.

**Patient and public involvement** Patients and/or the public were involved in the design, or conduct, or reporting, or dissemination plans of this research. Refer to the Methods section for further details.

**Patient consent for publication** Not required.

**Ethics approval** The study was approved by the University of Nottingham Business School Ethics Committee and all participants gave informed consent.

**Provenance and peer review** Not commissioned; externally peer reviewed.

**Data availability statement** Data are available upon reasonable request.

**ORCID iD**
Natasha Tyler http://orcid.org/0000-0001-8257-1090

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
