## [Reviewer comments · BMJ Open]

ARTICLE DETAILS

TITLE (PROVISIONAL)	Developing a Core Outcome Set for Interventions to Improve Discharge from Mental Health Inpatient Services: A Survey, Delphi and Consensus Meeting with Key Stakeholder Groups.
AUTHORS	Tyler, Natasha; Wright, Nicola; Grundy, Andrew; Waring, Justin

VERSION 1 – REVIEW

REVIEWER	Carmel Hughes Queen's University Belfast United Kingdom
REVIEW RETURNED	28-Oct-2019

GENERAL COMMENTS	Abstract: Line 21-results are reported in this part of the abstract which is not appropriate, as this should only report on the types of participants targeted for recruitment/participation rather than the numbers which took part. Numbers of participants should be reported in the Results section. Some typographical errors in the Strengths and Limitations section e.g. line 52/53 This is the first.. Introduction-sets the scene well Method Page 4 Line 46/47 insert 'a' for 'a systematic review...' Page 5, Line 3/4 For the online questionnaires... which questionnaires are being referred to at this point? Page 5, line 10/11. Twitter is mentioned as a platform for distribution, but I am not sure for what? Page 5, line 17/18. Reference is made to round 3. I assume that this refers to a Delphi round. There needs to be some earlier reference to the number of Delphi rounds undertaken for this later reference to make sense to the reader. Page 5, line 26/27-reference is made to 'the final round' but again, this is not clear. There needs to be some earlier overarching brief description of the methodology so that this makes sense. This section on participants is a mix of a description of the participants and methodology and is not very clear. It needs to be more stepwise in its approach to describing how participants were identified and recruited, with less emphasis on how the various stages of the development of the core outcome set proceeded. Stage 1-there needs to be a very clear description of how the qualitative questionnaire was undertaken. There are also some problematic sentences which are difficult to follow because of poor punctuation. E.g lines 37-39 Additional file 1.... My understanding is
--

	that if a stakeholder were a member of more than one group, they answered questions relevant to multiple groups? How was informed consent taken? The paragraph on page 5 which runs from line 34/35 to 49/50 covers administration, how participants were supposed to answer the question and the development of questions. The development section should come earlier in this paragraph as this stage had to be completed before distribution of the questionnaire. How were decisions made in relation to relative importance of outcomes? There are a couple of examples given so would it be true to state that outcomes that were excluded represented very 'niche' or 'specialised' areas of care? Page 6, line 11/12-14. There is reference to an outcome list, and instructions for questionnaires. Does this refer to the Delphi, as later in this para, the authors refer to round 1? Does the latter refer to the first round of the Delphi? These sentences are confusing. In relation to the Delphi, the authors report that it was conducted over two rounds (page 6, line 19/20). However, earlier on page 5, line 17/18, reference is made to round 3. Furthermore, the authors need to be consistent in the way in which they refer to the numbering of rounds, using either the figure (1 or 2) or words (one, two). By convention, it should be the figure. Again, the organisation of the text needs some attention. Reference to the way in which the survey was accessed (through a link sent to respondents' emails) needs to appear earlier in this paragraph than it currently does. Why did the authors opt for a 7 point scale and not a 9 point scale which is often used in COS development studies? For the consensus meeting, some of the sentences are very long and would benefit from punctuation. It is not clear from where participants were recruited? At the end of the paragraph (page 7, line 35/36), reference is made to a four item core outcome set. This is a result and should not be presented in the Method. Another stage is described on page 7 (Preliminary Measurement Recommendations). This was not reported in the abstract. Reference is made to 'three rounds' (page 7, line 41/42), so does this mean that participants were recruited from the Delphi panel? Page 7, line 44/45-I did not understand the line 'The invitation made it clear that the questionnaire is most relevant...'; This section is very confusing. Again, on line 48/49, reference is made to the four core outcomes. This represents results. There needs to be a definition of a time marker. Does this refer to when an outcome should be measured? Results The numbers of participants presented (n=93) does not seem to equate to the individual category numbers shown. The authors do state that respondents selected multiple categories, but this needs to be better described. Page 8, line 39/40-how did the authors know that new participants joined in Round 2? Did they approach new participants for the second round? Page 8, line 39/40-it is confusing when the authors refer to an attrition rate between the first questionnaire and Round 1. Is the first questionnaire relating to the qualitative questionnaire? I would see that latter as quite distinct from the Delphi. I found it difficult to follow the inclusion/exclusion of outcomes in the 2nd para of the Delphi process (page 8). The tables are not particularly helpful. Figure 1 covers everything, but it might be easier to break it up into a series of figures.
--	--

	Page 10, line 30/31-tremendous agreement?? What does this mean? Reference is also made to text that was added to the Discussion, but this does not seem to be appropriate to do at this point. Page 11, line 9/10-what does it mean that the core outcome set of four was ratified? Page 11, Table 2. Is there any significance in why some X/x are upper or lower case? Page 11, line 19/20 reference is made to Figure 2. I could not find Figure 2 The sections describing the various measures-page 13- are not well-written with long sentences and inappropriate punctuation. What is a 'popular' recommendation? The section 'rambles' and is not succinct, and although many recommendations are made in respect of 28 day data collection, the authors have also proposed many other options in terms of timing, so it does not appear to have been particularly well-considered. The standard approach for identifying methods of measurement is via COSMIN, but I don't believe that the authors have used or referred to this. I don't think this part is a particular strength of the paper. This section is quite long (all of page 13), but was not covered in the Abstract. Discussion Page 14, line 17/18-the authors refer to outcomes as being 'critically important', but is this how the respondents were asked to judge proposed outcomes? I thought the scale focused on agreement for inclusion. Later on this page (line 53/54), the authors refer to outcomes that are 'essential', but again, are these ways in which the respondents were asked to judge outcomes? Again, as stated before, reference is made to attrition rates between the questionnaire and Delphi survey (page 15, line 47/48), but I am not sure that this is a helpful point. There are a number of issues with the references, which need to be carefully checked and formatted-for example, see ref 6, where authors' names are a series of initials. This also applies to other references. Some journal titles are written entirely in capital letters e.g. see ref 4. This section needs to be very carefully checked as there are a number of other problematic citations.
--	---

REVIEWER	Molly Horstman Baylor College of Medicine, USA
REVIEW RETURNED	26-Nov-2019

GENERAL COMMENTS	This article describes the process through which the authors developed a set of core outcome measures for care transitions from inpatient mental health to the community. The authors present a clear rationale for this study and had an appropriate description of the methods used (Delphi method and consensus panel). One of the strengths of their process was the inclusion of end users at each stage of outcome measure set development. Major Comment: 1. In Table 1, I was initially surprised to see a high median for the measures that were included. On pg 6, the methods state that a Strongly Agree and Agree were 1-2 and Disagree and Strongly Disagree were 6-7 - however, later on the same page it says that Strongly Agree and Agree were 6-7. Which is the correct one? Minor Comments:
---

	1. Abstract: It is unclear what you mean by "groups" in the first sentence under Participants. Would improve clarity if you moved the third sentence ("Participants were from five stakeholder groups...") before the first sentence. 2. Overall: The manuscript is long and could be edited to reduce the length without impacting the message. For example, on pg 14, lines 19-25, have two sentences back to back that essentially say the same thing and on pg 15, line 3-8 could be deleted as this was covered in the results section. 3. On Pg 5, line 54, should it be line-by-line coding? I am not familiar with the phrase line-by-coding, but this may be the common terminology in other countries.
--	---

VERSION 1 – AUTHOR RESPONSE

Reviewer 1	
Line 21-results are reported in this part of the abstract which is not appropriate, as this should only report on the types of participants targeted for recruitment/participation rather than the numbers which took part. Numbers of participants should be reported in the Results section.	Thank you we have moved this to the results section of the abstract.
Some typographical errors in the Strengths and Limitations section e.g. line 52/53 This is the first..	Thanks we have removed these typographical errors.
Method Page 4 Line 46/47 insert 'a' for 'a systematic review...'	Thank you we have added this
Page 5, Line 3/4 For the online questionnaires... which questionnaires are being referred to at this point?	Thank you for noticing we have removed this reference
Page 5, line 10/11. Twitter is mentioned as a platform for distribution, but I am not sure for what?	I have changed the word distribution to recruitment. Thanks for noticing this error.
Page 5, line 17/18. Reference is made to round 3. I assume that this refers to a Delphi round. There needs to be some earlier reference to the number of Delphi rounds undertaken for this later reference to make sense to the reader.	I have removed the numbers from this sentence and instead described the iterative process that involves the same participant across multiple rounds. I think this reads much better, thanks for noticing. Please see first paragraph on page 5.
Page 5, line 26/27-reference is made to 'the final round' but again, this is not clear. There needs to be some earlier overarching brief description of the methodology so that this makes sense. This section on participants is a mix of a description of the participants and methodology and is not very clear. It needs to be more stepwise in its approach to describing how participants were identified and recruited, with less emphasis on how the various stages of the development of the core outcome set proceeded.	I have moved some of information that was in the participant section to more appropriate places in the paper. I have added the number of Delphi rounds into the overarching brief description of the methodology. Please see study overview section at the end of page 4.
Stage 1-there needs to be a very clear	We have added more detail into this section,

description of how the qualitative questionnaire was undertaken. There are also some problematic sentences which are difficult to follow because of poor punctuation. E.g lines 37-39	describing the qualitative questionnaire, thank you for pointing this out, the changes improve the paper. We have also changed the structure of this section to improve readability.
Additional file 1.... My understanding is that if a stakeholder were a member of more than one group, they answered questions relevant to multiple groups?	This is a much simpler paraphrase, I have changed this thank you.
How was informed consent taken?	Thank you we have added some more information to this section.
The paragraph on page 5 which runs from line 34/35 to 49/50 covers administration, how participants were supposed to answer the question and the development of questions. The development section should come earlier in this paragraph as this stage had to be completed before distribution of the questionnaire. How were decisions made in relation to relative importance of outcomes? There are a couple of examples given so would it be true to state that outcomes that were excluded represented very 'niche' or 'specialised' areas of care?	We have changed the order and structure of this section to: development, administration and analysis. Thanks for pointing this out. We have included an extra line in this paragraph to describe how decisions were made. We have changed the wording to include specialised areas of care, I think this makes the sentence clearer, thank you.
Page 6, line 11/12-14. There is reference to an outcome list, and instructions for questionnaires. Does this refer to the Delphi, as later in this para, the authors refer to round 1? Does the latter refer to the first round of the Delphi? These sentences are confusing	We have moved this to stage 2, and edited the sentences for clarity, thanks. We have also added a few lines to describe the Delphi process at the beginning of this section.
In relation to the Delphi, the authors report that it was conducted over two rounds (page 6, line 19/20). However, earlier on page 5, line 17/18, reference is made to round 3. Furthermore, the authors need to be consistent in the way in which they refer to the numbering of rounds, using either the figure (1 or 2) or words (one, two). By convention, it should be the figure.	Sorry this was a typo, initially the qualitative questionnaire was called round 1, meaning there were essentially 3 rounds. However, I missed this one instance of referring to round 3 when it changed over. It has now been removed. We have changed all of the round references to figures.
Again, the organisation of the text needs some attention. Reference to the way in which the survey was accessed (through a link sent to respondents' emails) needs to appear earlier in this paragraph than it currently does.	This reference is now at the start of the paragraph thanks.
Why did the authors opt for a 7 point scale and not a 9 point scale which is often used in COS development studies?	We have added a sentence to clarify this decision in the paper. Although GRADE Guidelines suggests 9 (Guyatt et al. 2010), it is by no means definitive and there's no evidence for 9 versus other scales ('Guideline developers may choose to rate outcomes numerically on a 1–9 scale'). We have seen that it is common in COS studies to use 9 points but in other psychology literature people have found 7 to be optimal. We selected 7 but could have selected 9 and don't think it has had a significant impact on our findings. I can see

	that this is an important area to clarify in future.
For the consensus meeting, some of the sentences are very long and would benefit from punctuation. It is not clear from where participants were recruited? At the end of the paragraph (page 7, line 35/36), reference is made to a four item core outcome set. This is a result and should not be presented in the	The participant recruitment for the consensus meeting is now described the participant section of the methods (pg 5). We have removed the number of items. We have changed the structure and punctuation of this paragraph to improve clarity.
Another stage is described on page 7 (Preliminary Measurement Recommendations). This was not reported in the abstract. Reference is made to 'three rounds' (page 7, line 41/42), so does this mean that participants were recruited from the Delphi panel?	We have removed this preliminary measure section from the paper, as you pointed out that it was weaker and not based on COSMIN, we agree but wanted to allow researchers to operationalise the core outcome set in its current form. But due to time and budgetary restraints it was not a distinct study.
Page 7, line 44/45-I did not understand the line 'The invitation made it clear that the questionnaire is most relevant...; This section is very confusing.	This section has now been removed from the paper
Again, on line 48/49, reference is made to the four core outcomes. This represents results. There needs to be a definition of a time marker. Does this refer to when an outcome should be measured?	We have removed this reference, thank you for pointing this out. We have removed stage 4 of the process from the paper.
Results The numbers of participants presented (n=93) does not seem to equate to the individual category numbers shown. The authors do state that respondents selected multiple categories, but this needs to be better described.	We have tried to describe this more clearly, participants did not fit into distinct stakeholder groups.
Page 8, line 39/40-how did the authors know that new participants joined in Round 2? Did they approach new participants for the second round?	We have added some further clarification, this is described in the methods also. These were individuals that were involved in the process, so completed the qualitative questionnaire but not round 1 of the Delphi.
Page 8, line 39/40-it is confusing when the authors refer to an attrition rate between the first questionnaire and Round 1. Is the first questionnaire relating to the qualitative questionnaire? I would see that latter as quite distinct from the Delphi. I found it difficult to follow the inclusion/exclusion of outcomes in the 2nd para of the Delphi process (page 8). The tables are not particularly helpful. Figure 1 covers everything, but it might be easier to break it up into a series of figures.	We have removed this attrition rate, thanks. For the inclusion/exclusion description, I have added parenthesis to include more detail, I hope this helps. Think you are referring to what we have named figure 2 here (there seems to be an issue with the system). We have chosen to keep this as one figure to show the whole process.
Page 10, line 30/31-tremendous agreement?? What does this mean? Reference is also made to text that was added to the Discussion, but this does not seem to be appropriate to do at this point.	Sorry for the confusion, we did not add text to the discussion. We discuss something in the discussion of this paper- I have now made that clearer. I have removed the phrase tremendous agreement and changed to discussion about as agreement was only

	measured through anonymous voting.
Page 11, line 9/10-what does it mean that the core outcome set of four was ratified?	We have changed the word from ratified to agreed throughout.
Page 11, Table 2. Is there any significance in why some X/x are upper or lower case?	There is no significance, these were overlooked somehow and have now been changed.
Page 11, line 19/20 reference is made to Figure 2. I could not find Figure 2	Figure 2 is not shown in the manuscript, but there is a placeholder and is was uploaded to scholarone. It is available in my author area to view/download, I'm not sure what the problem is.
The sections describing the various measures-page 13- are not well-written with long sentences and inappropriate punctuation. What is a 'popular' recommendation? The section 'rambles' and is not succinct, and although many recommendations are made in respect of 28 day data collection, the authors have also proposed many other options in terms of timing, so it does not appear to have been particularly well-considered. The standard approach for identifying methods of measurement is via COSMIN, but I don't believe that the authors have used or referred to this. I don't think this part is a particular strength of the paper. This section is quite long (all of page 13), but was not covered in the Abstract.	We have removed this preliminary measure section from the paper, as you pointed out that it was weaker and not based on COSMIN, we agree but wanted to allow researchers to operationalise the core outcome set in its current form. But due to time and budgetary restraints it was not a distinct study.
Discussion Page 14, line 17/18-the authors refer to outcomes as being 'critically important', but is this how the respondents were asked to judge proposed outcomes? I thought the scale focused on agreement for inclusion. Later on this page (line 53/54), the authors refer to outcomes that are 'essential', but again, are these ways in which the respondents were asked to judge outcomes?	Thanks for pointing this out, this was an issue about expression rather than what was used in the methods. We have changed both instances to be in line with the expression used in the methods.
Again, as stated before, reference is made to attrition rates between the questionnaire and Delphi survey (page 15, line 47/48), but I am not sure that this is a helpful point.	Thank you. We have deleted this reference.
There are a number of issues with the references, which need to be carefully checked and formatted-for example, see ref 6, where authors' names are a series of initials. This also applies to other references. Some journal titles are written entirely in capital letters e.g. see ref 4. This section needs to be very carefully checked as there are a number of other problematic citations	Thank you for pointing this out. It is an issue with the reference management software that has now been solved. Thanks,
Reviewer 2	
Major Comment: 1. In Table 1, I was initially surprised to see a high median for the measures that were included. On pg 6, the methods state that a	Thanks for pointing this out, it has now been changed, with agreement represented as higher scores. Thanks.

Strongly Agree and Agree were 1-2 and Disagree and Strongly Disagree were 6-7 - however, later on the same page it says that Strongly Agree and Agree were 6-7. Which is the correct one?	
Minor Comments: 1. Abstract: It is unclear what you mean by "groups" in the first sentence under Participants. Would improve clarity if you moved the third sentence ("Participants were from five stakeholder groups...") before the first sentence.	This structure has changed due to reviewer one's comments and reflects your comments. Thanks.
2. Overall: The manuscript is long and could be edited to reduce the length without impacting the message. For example, on pg 14, lines 19-25, have two sentences back to back that essentially say the same thing and on pg 15, line 3-8 could be deleted as this was covered in the results section.	We have deleted one sentence for each of the suggestions. Reviewer one raised concerns about the rambling of the 'measurement recommendation' sections which have now been removed completely. Therefore, the paper is now much shorter.
3. On Pg 5, line 54, should it be line-by-line coding? I am not familiar with the phrase line-by-coding, but this may be the common terminology in other countries.	Thanks for noticing, it should be line by line.

VERSION 2 – REVIEW

REVIEWER	Molly Horstman Baylor College of Medicine, USA
REVIEW RETURNED	27-Jan-2020

GENERAL COMMENTS	This manuscript offers a standardized set of outcome measures for interventions aimed at improving care transitions for adult mental health care. The use of a unified measure set would enhance our understanding of which interventions best support adults during these difficult transitions. The inclusion of researchers, clinicians, and end-users in the process is a strength. I appreciate the edits made by the authors. My recommendation still is that the manuscript should be shortened to make it easier to read. There are many areas that could be edited/deleted, but I provide a few examples here. On page 4, line 44-59, there is more detail than the reader needs. On page 10, lines 44-49 repeat what is listed in the table. On page 11, line 48-50, this is not a result - should be deleted or moved. On page 14, line 42-45, this is not needed here given the removal of the recommendations from earlier in the paper. On page 14-15, 47-24, paragraph is long and hard to read. On page 15, line 53-55 this repeats what is said two paragraphs above. There is misalignment on the likert scale for the Delphi - on page 7, strongly agree is rated as a 1, but listed as a 7 on page 8. There are also some grammatical changes that should be made
--

	throughout. A few examples are: on page 5, line 24-28 - this would be easier to read as two separate sentences. On page 5, line 52-54, there seems to be a word missing. On page 7, line 8, it seems like it should be "too" not "two". On page 9, line 56-57, "this individuals" needs to be edited.
--	---

VERSION 2 – AUTHOR RESPONSE

My recommendation still is that the manuscript should be shortened to make it easier to read. There are many areas that could be edited/deleted, but I provide a few examples here. On page 4, line 44-59, there is more detail than the reader needs. .	I have reduced this paragraph thank you for the suggestion.
On page 10, lines 44-49 repeat what is listed in the table	I have deleted this section thank you for the suggestion.
On page 11, line 48-50, this is not a result - should be deleted or moved.	Thank you I have deleted.
On page 14, line 42-45, this is not needed here given the removal of the recommendations from earlier in the paper.	I have rephrased this sentence.
. On page 14-15, 47-24, paragraph is long and hard to read.	Thank you for this suggestion, I have separated this paragraph, paraphrased and removed some phrases.
On page 15, line 53-55 this repeats what is said two paragraphs above.	I have deleted this thank you.
There is misalignment on the likert scale for the Delphi - on page 7, strongly agree is rated as a 1, but listed as a 7 on page 8.	Thank you I have corrected this error.
There are also some grammatical changes that should be made throughout. A few examples are: on page 5, line 24-28 - this would be easier to read as two separate sentences	Thank you I have made this change.
. On page 5, line 52-54, there seems to be a word missing. On page 7, line 8, it seems like it should be "too" not "two".	Thank you for noticing these.
On page 9, line 56-57, "this individuals" needs to be edited.	Thank changes have been made.